# ISBAR: A Handover Nursing Strategy in Emergency Departments, Scoping Review

**DOI:** 10.3390/healthcare12030399

**Published:** 2024-02-04

**Authors:** Veronica Chaica, Rita Marques, Patrícia Pontífice-Sousa

**Affiliations:** 1Institute of Health Sciences, Universidade Católica Portuguesa, 1649-023 Lisbon, Portugal; patriciaps@ics.lisboa.ucp.pt; 2Hospital Garcia de Orta, 2805-267 Almada, Portugal; 3CIIS-Center for Interdisciplinary Health Research, Universidade Católica Portuguesa, 1649-023 Lisbon, Portugal; ritamdmarques@gmail.com; 4Portuguese Red Cross Health School, 1300-125 Lisbon, Portugal

**Keywords:** ISBAR, handover, communication, nursing care, emergency department

## Abstract

The present work aims to map the available scientific evidence on the benefits of using the ISBAR tool in the nursing care of acutely ill adult patients’ handover in an emergency department context. To this end, a scoping review was conducted, according to the guidelines proposed by the Joanna Briggs Institute (JBI), to answer the following research question: “What are the benefits of using the ISBAR tool in the nursing care of acutely ill adult patients’ handover in an emergency department context?” The bibliographic search was carried out during August and September 2023 in the following electronic databases: CINAHL Complete; MEDLINE Complete; Cochrane Central Register of Controlled Trials; Cochrane Database of Systematic Reviews; and Cochrane Methodology Register. Only works published between 2013 and 2023 were deemed fit for inclusion. All the included studies (9) show that ISBAR methodology, as a standardized tool for transferring nursing care in the emergency service, allows for a safe, clear, and concise transition of nursing care. The benefits relate to patient and professional safety, continuity, and quality of care, as well as patient and professional comfort, with health gains.

## 1. Introduction

Care transition, especially in inter- and intra-hospital transfers, has become increasingly important, both nationally and internationally [1]. In such moments, responsibilities and information on patient care are transmitted between the involved professionals—a process known as “handover” or “handoff” [2]—to ensure care continuity and patient safety [1,2,3].

Many authors view handover procedures in emergency departments as high-risk situations with respect to the occurrence of clinical errors. Concerning patient safety, care transition moments are considered vulnerable events if their complexity increases the risk of errors associated with the transmission of information. Within an institution, this occurs, for example, when a patient is transferred to a different level of care or when shifts change [1,2,3]. Many authors state that faults in the communication process can result in numerous errors, which may jeopardize patient safety [4,5,6,7].

Over the years, patient safety has become a major concern worldwide. Its importance has been highlighted by various international scientific organizations. A report published by the Joint Commission Center for Transforming Healthcare reveals that communication errors often generate sentinel events [7]. Several healthcare organizations—including the World Health Organization (WHO), the Joint Commission, the Agency for Healthcare Research and Quality, and the Institute for Healthcare Improvement—recognize the communication between the involved professionals as an essential component of safe care, especially during the handover process when patient information is transmitted from one nurse to another [2,6].

The handover process can promote a comforting transition, as it should comprise the transmission of a holistic view of the patient, as well as joint care planning, with the collaboration of the patient/family and all the involved professionals. This requires acknowledgment of the patient’s condition and circumstances, aiming to facilitate care changes [8].

In an emergency department context, the handover process is described as complex and unpredictable due to patients’ instability and high turnover rate. These aspects, in themselves, increase the risk of adverse events. Additionally, such environments demand intense care monitoring, rapid decision-making, and the involvement of multiple healthcare professionals [2,7]. While trying to ensure an immediate, efficient, and technically complex response, the focus is on the patient’s treatment. Accordingly, such settings seldom offer appropriate conditions for welcoming patients, respecting their privacy and individuality, engaging in therapeutic interactions, or sharing timely and adequate information. These aspects must be addressed to provide humanized care, which is safe and comforting for the hospitalized patient and their family [9]. Given the complexity mentioned above, comfort emerges as a need. Thus, when providing care to critically ill patients, nurses should view comfort as a purpose and as the goal of intentional interventions [10].

It is important to emphasize that, in a logic of comprehensive, person-centered care, ISBAR as a handover nursing strategy in emergency departments makes the action of caring a reality and facilitates continuity of care through comfort intentionality, which is of particular importance. In emergency settings, care is focused on the acutely ill patient who is experiencing specific health/disease and hospitalization circumstances characterized by uncertainty and vulnerability. The individual finds himself/herself in a context of dependence due to a lack/loss of autonomy in different dimensions (physical, psychological, or intellectual), requiring help and/or assistance, which becomes a need. Despite being intrinsic to the human condition, frailty and vulnerability are amplified in situations of acute illness and hospitalization by the patients’ limited possibilities and capabilities [11]. The individualization and humanization of care derive from the nurses’ attitudes during their professional activities. In an evolving dimension, they try to meet the patient’s needs, ensuring adequate care provision to relieve the patient’s suffering while also preventing complications, discomfort, and regression. Thus, the nurses’ purpose is to promote states of comfort [11].

In such circumstances, effective communication—by means of standardized tools—plays a vital role among emergency teams, as it encourages a safety culture and contributes to a successful care transition [12].

With respect to care transition, there are many applicable techniques. However, national and international regulatory authorities, as well as health service quality committees, recommend using the ISBAR tool (I—Identification; S—Situation; B—Background; A—Assessment; R—Recommendation). They consider it the most appropriate, structured, and standardized instrument for care transition moments. The ISBAR methodology can be broadly employed not only in emergency departments but also in hospital wards and pre-hospital services [7,12]. Research has shown that its use promotes interdisciplinary teamwork among healthcare professionals, also favoring patient safety and comfort [8,13]. From the professionals’ point of view, ISBAR facilitates articulation/discussion, thus allowing the creation of joint care plans comprising the chosen approach and the procedures to be carried out, consequently improving patient satisfaction and the results of decision-making associated with the care plan. At the same time, ISBAR helps resolve conflicts within the multidisciplinary team [9,12]. Therefore, it has been implemented in numerous healthcare systems worldwide [3].

In view of this reality, we decided to conduct a scoping review based on the guidelines proposed by the Joanna Briggs Institute (JBI) [14] to map the existing scientific knowledge on the benefits of using the ISBAR tool in the nursing care of acutely ill adult patients’ handover in an emergency department context.

## 2. Materials and Methods

The present study is a scoping review carried out in accordance with the methodology recommended by the JBI [14,15,16,17]. Employing the PCC system—with “Population” being acutely ill adult patients, “Concept” being the benefits of using the ISBAR methodology for patient care handover, and “Context” being emergency departments—we established the following research question: “What are the benefits of using the ISBAR tool in the nursing care of acutely ill adult patients’ handover in an emergency department context?” As for the search strategy, it was carried out during August and September 2023 in three stages: the initial search was limited to CINAHL and Pubmed to identify articles on the subject using the keywords: “ISBAR”, “Transition”, “Communication”, “Emergency department”, and “Handover”. We consider it important to know the benefits of the ISBAR methodology in the transition of nursing care in the Emergency Department context. The titles, abstracts, and indexing terms used to describe the articles considered relevant to this review were analyzed. In the second phase of the search, the keywords and terms identified were used to carry out a complete search in the databases included in the research for this study: CINAHL complete, MEDLINE complete, Cochrane Central Register Of Controlled Trials, Cochrane Database of Systematic Reviews, and Cochrane Methodology Register, which were adapted to each of them, such as using the Medical Subject Headings (MeSH) in MEDLINE, and combined with the Boolean operator “AND”. The inclusion criteria were full-text articles published in the last ten years (2013 to 2023), the purpose of which was to find out about the most recent scientific evidence, acutely ill patients aged between 18 and 64, written in Portuguese and English. Studies that included people under the age of 18, pregnant women, people with psychiatric disorders, and people in palliative care were excluded as they were not part of the focus of this review. To systematize the process of including studies, the Preferred Reposting Items for Systematic Reviews and Meta-Analysis (PRISMA) methodology was used [18].

Concerning the search strategy, the bibliographic search was performed during August and September 2023 through the EBSCOhost—Research Databases interface. From the available electronic databases, we selected the following: CINAHL Complete, MEDLINE Complete, Cochrane Central Register of Controlled Trials, Cochrane Database of Systematic Reviews, and Cochrane Methodology Register. To conduct the search, we used the MeSH descriptors “ISBAR”, “Transition”, “Communication”, “Emergency department”, and “Handover”, combined with the Boolean operator “AND”. At this stage, the following inclusion criteria were defined: works available in full text, published during the last ten years (between 2013 and 2023)—since we aimed to gather the most recent scientific evidence—which portrayed studies carried out with adult patients (between 18 and 64 years old), and written in Portuguese or English. To systematize the inclusion process, we employed the “Preferred Reporting Items for Systematic Reviews and Meta-Analysis” (PRISMA) methodology [18].

The obtained articles were pre-selected through title and abstract analysis after duplicate removal. To select the final sample, the works were read in full, and the previously established inclusion criteria were applied. This process resulted in the exclusion of repeated articles, as well as those that did not explore the benefits of using the ISBAR technique for care provision in an emergency department context. Both the search and the selection were performed by two independent reviewers. Whenever disagreements occurred, the opinion of a third reviewer was requested.

Subsequently, the studies were categorized according to their level of evidence and grade of recommendation based on the JBI guidelines [15]. This allowed a preliminary assessment of the studies’ methodological quality and of the evidence’s rigor [15,16,17].

To evaluate the studies’ methodological quality, we employed the JBI tool “Checklist for Diagnostic Test Accuracy Studies” [15]. Two independent reviewers carried out the data’s critical appraisal, extraction, and synthesis. Whenever disagreements occurred, the opinion of a third reviewer was requested. At this stage, there were no exclusions since all the considered studies presented high quality.

The included works underwent a critical analysis, with their results being interpreted and evaluated. Throughout this process, the key findings that answered the abovementioned research question were highlighted.

## 3. Results

The bibliographic search produced a total of 113 articles, 67 of which were excluded during duplicate removal (30) and title reading (37). Abstract reading was carried out on the remaining 46 works, of which 17 were approved for full-text reading and inclusion criteria application. Of those, only nine were included in the final sample, as shown in Figure 1.

The results of analysis of the included works were systematized employing an instrument adapted from the Methodological Manual for Scoping Reviews of the Joanna Briggs Institute [14,15,16,17]. Table 1 shows the outcome with respect to authors, year of publication, research design, studied population, and level of evidence.

As can be noticed in Table 1, the present scoping review encompassed one level 2.c study (which followed a quasi-experimental controlled prospective methodology) [2], three level 4.b studies (which followed an observational descriptive cross-sectional methodology) [3,19,20], and five level 2.d studies (which followed a pre-test/post-test or historical/retrospective control group methodology) [7,21,22,23,24]. Table 2 summarizes the outcome with respect to authors, objective(s), and main conclusions.

## 4. Discussion

By analyzing the results obtained, we found that regarding the benefits of the ISBAR methodology in the handover of nursing care in the emergency department, research efforts are scattered over time. 

Furthermore, the included works are mainly quantitative in nature. As far as the studied populations are concerned, they are mostly groups of healthcare professionals, which include nurses and doctors. However, this review was restricted to publications referring only to nurses.

Scientific evidence demonstrates that the handover process in an emergency department context has been increasingly valued. In this sense, patient transfers have been internationally recognized as an area of risk with respect to patient safety. The care transition procedures carried out in such environments are highly variable and are performed under unstable conditions. Accordingly, the available literature argues that they should be structured using standardized tools, in particular the ISBAR, to reduce errors and minimize the occurrence of adverse events [2,19,20,21].

The benefits of ISBAR in the handover of nursing care in the emergency department identified in the studies relate to (i) Patient and professional safety; (ii) Continuity and i86755In this field, this methodology facilitates the development of praxis at a technical level. About the continuity and quality of care (ii), there are benefits related to saving time and standardizing care [2], the clear and concise transmission of information [2], the loss of information about the patient [7], the development of critical thinking [3,7,23,25], and improving the quality and effectiveness of the transition of care [21]. Finally, this methodology appears to facilitate patient and professional comfort (iii) as it facilitates execution [2], professional responsibility [2,7], and increased trust and collaboration between professionals [23]. These domains facilitate the development of praxis in terms of technical, human, relational, and ethical skills, which allows decisions to be taken in accordance with the situation. 

However, there are benefits that are linked to the different dimensions and, in themselves, interconnect and intersect, giving the benefits of ISBAR a transversal character, namely effective communication [3,7,24], the quality of the transition of care between the different teams [19], and the standardization of the information transmitted [2]. It is often classified as accessible [2,7], allowing a clear, concise, and simple communication process [2,3,20,26] while also preventing significant losses of information [7]. In addition, the use of ISBAR during handover procedures seems to improve the multidisciplinary team’s engagement and the reliability of communication. 

The complexity of the handover in the emergency department, therefore, requires this structured methodology, as a lack of standardization can lead to high variability in the information transmitted, a deficit in the reassignment of professional responsibilities, a delay in medical diagnosis, the occurrence of adverse events, and/or ineffective/wrong treatment [20]. All these negative consequences stem from unproductive communication between the professionals involved [22].

Moreover, communication problems can originate a lack of trust between the care recipient and the caregiver, often instigating situations of conflict. This results in wasted time and fragmentation of the processes that involve the multidisciplinary team, jeopardizing the patient’s safety, quality, and comfort [2,19,20,21,22].

In end-of-life care situations, which generate anxiety, discomfort, and distress, the ISBAR methodology facilitates the structuring of information [24] to promote the continuity of information, as well as the comfort and well-being of the patient and family [27]. In this way, handover is based on a humanizing relationship, where the nurse’s attention, sensitivity, availability, and concern are transmitted through communication skills promoted through ISBAR. This should be carried out at the patient’s bedside so that they can integrate their own care plan and be empowered with up-to-date clinical information provided by the nurse [10]. 

ISBAR, due to its versatility, simplicity, and adaptability to different clinical domains [2,19], promotes nurses’ situational awareness, which contributes to the development of critical thinking and decision-making [3,7].

The employment of standardized communication tools to perform the handover process is generally well received by the multidisciplinary teams’ professionals, namely in emergency departments. As such, the usefulness of the ISBAR technique is a transversal and unanimous finding throughout the analyzed literature [2,7,19,27]. One of the included studies reports a 35.7% receptiveness concerning the application of ISBAR by the emergency department’s nursing staff [2].

Furthermore, some of the included studies claim that when the handover process is carried out employing the ISBAR tool, there is a greater adherence of the whole multidisciplinary team with respect to the reassignment of professional responsibilities [2,7,19]. This ultimately leads to improvements in quality, safety, and comfort indicators, which benefit the patient [19].

All the included literature affirms that communication by means of ISBAR is the most consensual method for transmitting information in an emergency department context. Notwithstanding, two of the included works concluded that the effectiveness and efficiency of the tool’s application were influenced by professional training. Emergency department teams are generally aware of the existence of ISBAR and other standardized instruments, but they seldom employ those resources of their own accord. However, when their training addresses the importance of standardized communication, they are the first to acknowledge the relevance of using such tools [3,19].

Complementarily, the results obtained highlighted four main aspects that underpin the usefulness of ISBAR as a tool for interprofessional and standardized communication: the use of a common language in interdisciplinary communication, eliminating language barriers; efficient organization of the information transmitted; facilitation of collaborative team-based communication (including conflict resolution and shared decision-making); and, finally, versatility (since it can be applied in different contexts, such as face-to-face discussions, group presentations, email communication, and drafting approval documents). 

Research shows that ISBAR is a standardized, valid, and effective communication tool, both from the point of view of patients and professionals, and is recognized by the Joint Commission, the Agency for Healthcare Research and Quality, the Institute for Health Care Improvement, and the WHO [6].

Within the scope of this scoping review, we analyzed nine studies that made it possible to answer the research question. Thus, we can affirm that the use of ISBAR for the transfer of care in emergency services is beneficial, as can be seen from the various advantages identified in the available literature.

The main limitations of this study stem from the scarcity of relevant literature available regarding the application of ISBAR in the Handover of nursing care in the Emergency Department. The literature on this subject focuses mainly on hospital inpatient settings. The fact that most articles have a level of evidence of 2d and 4b may constitute a limitation in the findings. The scarcity of nursing literature on this subject reinforces the need for further research.

## 5. Conclusions

This scoping review allowed us to identify the benefits of using the ISBAR methodology as a standardized tool for transferring nursing care in the emergency service. The benefits relate to patient and professional safety, continuity and quality of care, and patient and professional comfort.

The ISBAR is considered an option for the handover of nursing care, suitable for emergency services due to its structural simplicity, universal language, and the possibility of adapting it to different clinical domains. 

Scientific evidence also demonstrates that its use benefits multidisciplinary teams and the people cared for, with a strong positive impact on safety during clinical transitions. The use of ISBAR increases awareness of the importance of adopting structured and effective communication, as well as the need for structured knowledge about patient care, allowing an individualized, safe, and comforting response with health gains.

The need for further research on this topic derives from the importance of evidence-based practice. Accordingly, the subject under study should continue to be discussed and debated by the scientific community.

## Figures and Tables

**Figure 1 healthcare-12-00399-f001:**
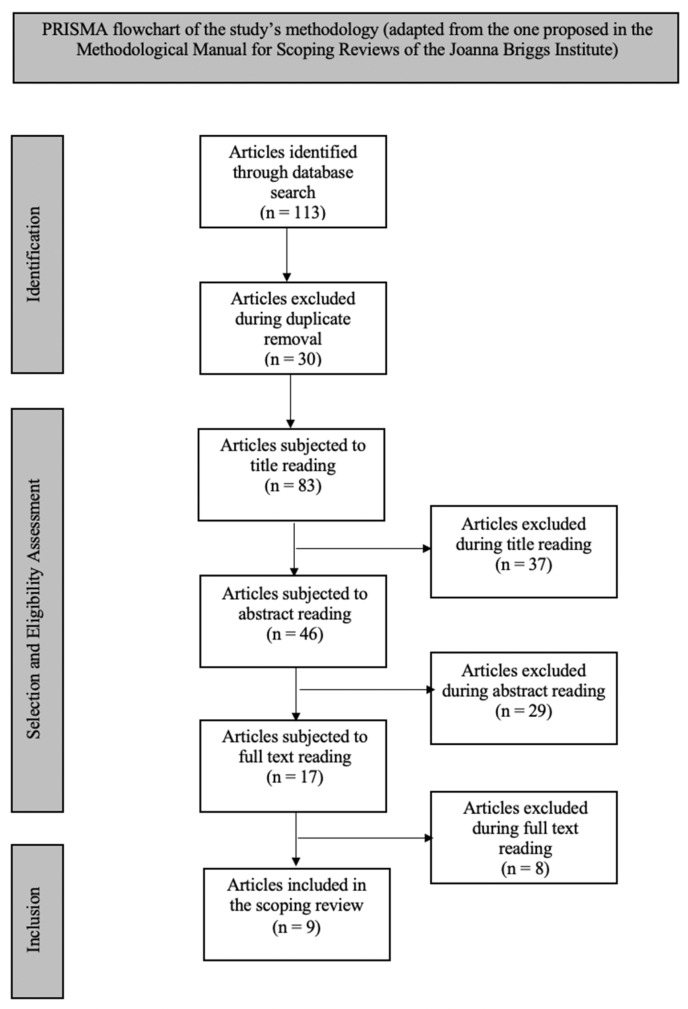
PRISMA flowchart of this study’s methodology (adapted from the one proposed in the Methodological Manual for Scoping Reviews of the Joanna Briggs Institute).

**Table 1 healthcare-12-00399-t001:** Included works organized by authors, year of publication, research design, studied population, and level of evidence.

Authors	Year of Publication	Research Design	Studied Population	Level of Evidence
Bakona, S.; Millichampb, T. [2]	2017	Quasi-experimental controlled prospective study	Nurses	2.c
Castro, C.; Marques, M.; Vaz, C. [3]	2022	Observational descriptive cross-sectional study	Nurses	4.b
Marmor, G.; Yonhong, M. [19]	2017	Observational descriptive cross-sectional study	Physicians	4.b
Ehlers, P.; Seidel, M.; Schacher, S.; Pin, M.; Fimmers, R.; Kogej, M.; Graff, I. [20]	2020	Observational descriptive cross-sectional study	Physicians	4.b
Campbell, D.; Dontje, K. [7]	2018	Quasi-experimental pre-test/post-test study	Nurses	2.d
Yegane, S.; Shahrami, A.; Hatamabadi, H.; Zijoud, S. [21]	2017	Quasi-experimental pre-test/post-test study	Physicians and nurses	2.d
Dojmi, F.; Mancini, N.; Nota, T.; Pisanelli, P. [22]	2015	Quasi-experimental pre-test/post-test study	Physicians and nurses	2.d
Meester, K.; Verspuy, M.; Monsieurs, K.; Bogaert, P. [23]	2013	Quasi-experimental pre-test/post-test study	Nurses	2.d
Dahlquist, R.; Reyner, K.; Robinson, R.; Farzad, A.; Laureano-Phillips, J.; Garrett, J.; Young, J.; Zenarosa, N.; Wang, H. [24]	2018	Quasi-experimental pre-test/post-test study	Physicians and nurses	2.d

**Table 2 healthcare-12-00399-t002:** Included works organized by authors, objective(s), and main conclusions.

Authors	Objective(s)	Main Conclusions
Bakona, S.; Millichampb, T. [2]	To improve the handover’s consistency through the development and application of a structured form designed for the transition from the emergency department to the ward.	The form’s application was well received by the emergency department’s nursing staff (35.7%). Most of the participants (64%) reported that the form’s application presented the following benefits: it saved time; it was easy to perform; it allowed a clear and concise transmission of information (60%); it promoted universality throughout the different services (79.1%); it encouraged professional responsibility.
Castro, C.; Marques, M.; Vaz, C. [3]	To ascertain the nurses’ opinions on care transition during shift changes in the emergency department and to appraise the nursing staff’s knowledge regarding patient safety.	As a standardized tool, the ISBAR methodology contributes to decision-making and critical thinking while allowing effective communication. This instrument has played a prominent role in care provision, being implemented in numerous healthcare systems worldwide. The professionals’ training and the use of standardized tools are essential strategies to ensure patient safety during care transition—a basic principle of health care and of the nursing practice.
Marmor, G.; Yonhong, M. [19]	To develop and apply a model for improving shift change practices by increasing the reliability of communication.Also, regarding this process, to identify possible negative effects on patient care and to assess the team’s adherence and acceptance.	The ISBAR tool was considered simple, being well accepted by the staff, and its application improved the quality-of-care transition between different teams. With respect to care transition during shift changes, standardized communication achieved through the ISBAR tool increased the team’s engagement and the communication’s reliability, thus improving patient safety. However, it did not enhance the quality of the documentation delivered for the patient’s medical record.
Ehlers, P.; Seidel, M.; Schacher, S.; Pin, M.; Fimmers, R.; Kogej, M.; Graff, I. [20]	To analyze the current practices regarding care transition in the emergency department, focusing on the application of a standardized tool (with specific content, purpose, and structure).	Regarding the care transition process, a lack of standardization leads to high variability in the transmitted information and to a deficit in the reassignment of patient care responsibilities. Protocols help to homogenize the handover procedures while providing guidance on the information that should be transmitted.
Campbell, D.; Dontje, K. [7]	To effectively implement the nursing care transition process derived from shift changes in the emergency department, also evaluating its impact.	The ISBAR tool is easy to use and avoids the loss of patient information, allowing better communication. Through adequate handover, nurses can promote situational awareness by observing care transition “on the spot”. This allows discussing the most appropriate care plan for the patient in question, ultimately facilitating the reassignment of responsibilities among the involved nurses.
Yegane, S.; Shahrami, A.; Hatamabadi, H.; Zijoud, S. [21]	To audit current clinical handover procedures, based on the ISBAR tool and to assess the effects of ISBAR training among the emergency department’s staff with respect to the improvement of patient transfers.	Emergency department teams should use a standardized instrument during clinical handover to improve the quality and effectiveness of care transition, as well as to reduce the number of adverse events resulting from ineffective communication. The emergency department’s staff needs to acquire care transition skills through adequate training to ensure patient safety. Through such training, professionals will be able to understand and perfect the ISBAR tool’s use while becoming aware of the importance of safe communication.
Dojmi, F.; Mancini, N.; Nota, T.; Pisanelli, P. [22]	To evaluate the communication process that occurs during the clinical transition between the pre-hospital team and the emergency department team using realistic scenarios.	Professional training on standardized tools for care transition was considered relevant by the studied population. Mechanisms such as standardized tools (e.g., ISBAR) are generally viewed as strategies to reduce the assessment errors made by emergency department healthcare professionals.
Meester, K.; Verspuy, M.; Monsieurs, K.; Bogaert, P. [23]	To determine the effects of employing the ISBAR communication tool with respect to the incidence of serious adverse events in hospital wards.	The application of the ISBAR instrument significantly reduces unexpected deaths. It also allows nurses to acquire critical thinking skills, thus enhancing their confidence regarding the patients’ assessment. After the introduction of ISBAR, the communication process improved, and the collaboration between the involved professionals became more effective.
Dahlquist, R.; Reyner, K.; Robinson, R.; Farzad, A.; Laureano-Phillips, J.; Garrett, J.; Young, J.; Zenarosa, N.; Wang, H. [24]	To ascertain if implementing a bedside patient handoff by means of standardized tools and reporting systems improves the performance and outcome associated with patient care.	Carrying out handoff procedures at the patient’s bedside, along with a standardized care transition using the ISBAR tool, improved the communication process, which became concise, effective, and time-efficient. Despite the wide range of available standardized instruments, ISBAR is the generally preferred tool, being recommended and validated using the studied population.

## Data Availability

Requests for research data may be addressed to Veronica Chaica.

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
