# Peer review of "ISBAR: A Handover Nursing Strategy in Emergency Departments, Scoping Review"

_healthcare, 2024, doi:10.3390/healthcare12030399_

Round 1
Reviewer 1 Report
Comments and Suggestions for Authors
This paper is a scoping review of the benefits of using ISBAR for patient handover in emergency departments. It is a worthy area of investigation due to the impact of handover on multiple aspects of care, importantly patient safety. Although the abstract states the research question is focussed around the benefits to patient care of ISBAR use in the emergency department, this isn’t fully addressed and many other themes appear to be introduced that aren’t totally linked to this question.
The introduction provides a general background to the topic, however it is confusing as to whether the paper is focussing on nurses’ use of ISBAR or the use of ISBAR in general. There is no reference in the title or abstract that nurses are the main focus but they are then singled out (throughout the paper). Please clarify this – either change the title or include discussion of other health professionals.
Comfort is also raised in the introduction (lines 65-89) but seems very out-of-context. I’m not sure exactly what point was trying to be raised and it also doesn’t link to any of the results, tables, or discussions. This should either be removed or directly linked to the results/discussion.
The paragraph beginning line 90 appears to repeat themes raised earlier in the section. Is there a specific point that is being highlighted? If so, re-writing/re-wording may emphasise this. References are also needed relating to line 91.
The general rationale for conducting the review needs some strengthening.
Materials and Methods are generally clear. It would be helpful to explicitly state if there were no exclusion criteria. Some questions regarding the search strategy – how were the original search terms identified? Was there any checking for additional search terms following the initial identification of papers? Were all 5 search terms needed concurrently, i.e. was “transition” needed as one of the 5 search terms for a paper to be returned, or would a paper be identified using only “ISBAR” and “Communication” and “Emergency Department”? This may have limited the number of papers so please clarify (it sounds like a combination of all 5 were needed from the methods).
The results could include more detail on the alignment to general themes related to the rationale for the review. Clarifying the purpose of the review should help to improve the results to ensure the important topics/points are highlighted. The general layout of the included tables is good (some random Portuguese below Table 1 needs removing).
The discussion is very repetitive and doesn’t adequately synthesise or explain the results. There is limited discussion around the main themes identified in the included papers – much of this section is very similar to the introduction. The section needs re-writing to integrate and develop ideas from the papers to answer the research question. There are also more comments specifically related to nurses, however this hasn’t been specified anywhere as a focus of the review.
I am unsure as to the comments “… they are mostly groups of health professionals” (lines 185-186). Who else was included that didn’t fit this criterion? Only health professionals were included in Table 1.
The sentence “As a result, the patient should be able to develop a life project, being empowered through updated clinical information provided by the nurse” (lines 223-225) has multiple potential interpretations, possibly a translation issue. This paragraph again appears to move outside the suggested scope of the review and doesn’t align with the results.
Comments on the limitations of the review don’t match the information in Table 1 (most of the included papers involved nurses – fewer studies involved physicians).
The conclusions would be improved by clearly addressing the research question in the context of the rationale – this section needs re-working once the rationale is clarified. The statement “ISBAR is considered the most appropriate option…” in the conclusion has not been referenced previously – there is also no mention of other tools (only that they exist). This should be introduced and discussed earlier.
Reference list: some references are a mix of Portuguese and English within the one reference (e.g. ref 5). Please check references are listed in their published language.
Figure 1 is in Portuguese, not English.
Throughout the paper gender is dealt with as binary (e.g. line 61 “his/her”). Please change these to non-gendered pronouns (e.g. “their”).
Some of the issues identified above may be related to translation into English. There are multiple grammatical and spelling errors that may have resulted in an incorrect interpretation of a word/sentence. Please check that the original intent is correct following translation.
Comments on the Quality of English LanguageSome of the issues identified above may be related to translation into English. There are multiple grammatical and spelling errors throughout the paper that may have resulted in an incorrect interpretation of a word/sentence. Please check that the original intent is correct following translation.
Author Response
Good Morning, Dear Reviewer The working group is very grateful for your review and contribution to our study, as we know that through it we had the opportunity to improve it. Please find attached the response to the revision proposals requested by you. Best regards, Verónica Chaíça

Reviewer 2 Report
Comments and Suggestions for Authors
Thank you for the opportunity to review a manuscript that addresses an important issue of patient safety in the emergency department. It emphasises well that precise and clear communication is one of the most important prerequisites for reducing avoidable adverse events. There are only a few comments to improve the article for publication.
1. In Figure 1, the text needs to be translated from Portuguese into English.
2. Please check the numbering of the included studies, as some of them do not correspond to those in the references.
3. The second to sixth sections of the discussion belong thematically more in the introduction and furthermore could be shortened.
Author Response

(The authors gave the same response as above.)

Reviewer 3 Report
Comments and Suggestions for Authors
thankyou for the opportunity to review your manuscript
abstract. concise
introduction. concise summary of the background to handover in ED. although there is some mention of why it is important to get right, it would be better to understand the 'degree' of importance - how much harm is actually being reported from poor handover? what is the current standard that we are trying to improve?
materials and methods. standard
results. figure 1 not in english, otherwise well laid out
discussion. a nice summary of the results. this paper focuses specifically on emergency departments yet there is no discussion about this context. the limitations should be expanded to discuss the level of evidence of papers found (mostly 2d and 4b)
conclusion. again concise.
Comments on the Quality of English Language
some gramatical areas throughout. figure 1 is not in english
Author Response

(The authors gave the same response as above.)

Round 2
Reviewer 1 Report
Comments and Suggestions for Authors
I don't believe there is an adequate rationale for the scoping review - why have they chosen to undertake the review? What are they hoping to add to the field by completing it? There are similar reviews that have been undertaken and I can't see any reason for this one (there may be one, but it hasn't been explained). I had requested a clear justification for the review in the revisions, but this wasn’t provided. There are also a number of papers included in the other previously-published reviews that have been excluded in this paper. I was hoping the revision may better explain why these papers would have been excluded when the methods were revised, but there doesn't appear to be a valid reason. This is also related back to the lack of rationale for the scoping review in the first place - without the clear question trying to be answered, it is difficult to tell if it is something that can be easily addressed or is a fundamental flaw in the paper. The rationale issue also impacts the results, with some comments/discussion around the results not aligning with various sections throughout the paper. This issue has not been addressed at all in their revisions. The authors specifically exclude papers discussing end-of-life care in their methods, yet then discuss papers that have focussed on end-of-life care and use these as justification for parts of the discussion. A clear research question may help their focus.
As the manuscript currently standards, it is repeating previous work but with unexplained gaps and therefore is not at the level of papers that have been published elsewhere. A valid and robust justification for undertaking a scoping review may explain the purpose and would then allow for minor revisions but, without this, I believe the manuscript should be rejected.
Author Response
Dear Reviewer,
The authors of the manuscript entitled, "ISBAR: A Handover Nursing Strategy in Emergency Departments, Scoping Review", thank you in advance for the opportunity to answer the questions raised by the reviewer.
